# Drivers of Consumer Participation in Online Second-Hand Transactions [†]

**Pedro Hinojo [1,2,*], David Suárez [3] and Begoña García-Mariñoso [3]**

[1] Department of Economics and Business, Universidad de Alcalá, 28801 Alcalá de Henares, Spain

[2] Department of Information Society Services, Comisión Nacional de los Mercados y la Competencia (CNMC), 08018 Barcelona, Spain

[3] Department of Statistics and Knowledge Management, Comisión Nacional de los Mercados y la Competencia (CNMC), 08018 Barcelona, Spain; david.suarez@cnmc.es (D.S.); begona.garcia@cnmc.es (B.G.-M.)

[*] Correspondence: pedro.hinojo@cnmc.es or pedrohinojo@gmail.com; Tel.: +34-936036191

[†] Disclaimer: The opinions and analysis that appear in this paper are responsibility of the authors only and do not necessarily represent those of the CNMC.

**Abstract:** Consumer participation in second-hand transactions is increasing, facilitated by digital platforms in the form of apps or websites. This study sheds light on the factors behind consumers' decisions to demand used goods via online platforms. Applying a logit model to a sample of 6705 internet users in Spain, we explore the role of economic variables, situational factors and individual characteristics. Our original findings indicate that the use of online platforms to buy or rent second-hand goods is more likely when being male, relatively young, with children, a frequent internet user, with employment and living in a household with some price-consciousness and environmental awareness. The scarcity of brick-and-mortar stores in the area and car ownership can also increase demand for used goods through online platforms.

**Keywords:** second-hand shopping; P2P consumption; collaborative consumption

## 1. Introduction

The use of online apps or websites to buy or rent goods has increased remarkably during last years. For instance, in Spain, the most popular digital app for carrying out second-hand transactions, Wallapop [1], has been among the top ten most downloaded in specific years, outstripping Instagram or Spotify [2]. As an illustrative example of the importance of the phenomenon, the global market of second-hand apparel is likely to expand from $28 billion in 2019 to $64 billion in 2024, which implies an outstanding 18% annual growth rate [3]. Similarly, the forecasts for the second-hand furniture market foresee the market share for the online stores segment to grow at a 10.4% annual rate [4].

However, despite its development, this phenomenon is relatively unexplored since adequate data to draw empirically based conclusions is lacking. Many authors emphasize the lack of empirical research on the use of second-hand peer-to-peer (P2P) platforms [5] and the fact that knowledge of second-hand goods online shopping is still in its infancy [6]. Thus, we overcome this difficulty by using an appropriate sample to contribute to the literature by explaining the drivers of consumers' decisions to buy or rent used goods by using internet apps/websites.

We also contribute to the literature on the wider phenomenon of "Collaborative Consumption" (CC), which has been defined as "the peer-to-peer-based activity of obtaining, distributing, sharing or giving the access to goods and services, coordinated through community-based online services, normally for a fee or other type of compensation" [7,8].

Two underlying ideas seem prominent in the disruptive changes that collaborative consumption brings about. First, the use of (otherwise) under-utilized resources to create

value for individual owners and second, the real novelty: the use of digital platforms to match transactions between peers at a very large scale [9,10].

Indeed, second-hand markets have existed for many years in advance to digitization, with no or little third-party intermediation; for example, in the form of flea markets, car boot or garage sales, or through one's network of individual contacts. However, digital platforms generate network effects/externalities [11], increase connectivity among a larger pool of agents (buyers and sellers), and facilitate a more efficient matching by reducing transaction and bringing-to-market costs [12]. Moreover, digital tools allow for the building of user reputation (through reviews, ratings, and recommendations), which helps buyers and sellers to establish trust between unknowns [13,14]. This is critical in online second-hand transactions [15], since mistrust is one reason why some consumers may prefer personal sources to apps/websites [14,16].

This work follows the stream of the literature which models factors behind participation in CC [8,17–21] and in the specific area of second-hand online consumption [5,6,10,13,22,23]. In particular, it contributes to the knowledge of the drivers of consumer participation in online second-hand markets by addressing the following gaps in the literature:

- First, we use a large and appropriate sample to understand the phenomenon. Our analysis is based on a representative sample of more than 6700 individuals of the Spanish population, including users and non-users of "second-hand" online platforms. This is considered of utmost importance for research [10,19,24], since most of the previous literature is based on samples of platform users or even qualitative research based on small "focus groups," making it difficult to empirically assess the determinants of the decision to engage in the collaborative economy.

- Second, we consider the actual use of platforms to engage in second-hand transactions rather than relying on stated preference, willingness, or intention to participate [10,25], as the latter does not necessarily imply actual engagement.

- Third, we include situational factors [26,27] related to context-specific variables such as the number of retail stores in the vicinity of the individual, from which policy implications can be derived. Previous literature [28] points out that the lack of understanding of such drivers of CC is an explanation for the lack of consistent policy recommendations. The role of interventions and situational factors has been explored in specific areas of the sharing economy and CC like carpooling [29], but not in the case of online second-hand transactions.

- Fourth, we explore the role of drivers of online second-hand shopping that relate to the multifaceted concept of sustainability and conclude by discussing what our results indicate about the impact of online second-hand transactions on economic, social, and environmental sustainability. From the economic standpoint, we investigate the influence of consumers' income and price awareness. From a social perspective, we assess the impact of socio-demographic factors (like age, gender, and the fact of having children) and variables like the familiarity with the internet. From an environmental viewpoint, we examine the effect of consumers' environmental awareness, together with other variables that have an environmental dimension (like car ownership).

Our findings are of significance to retailers and policy makers and also have academic implications. The results characterize the participants in online second-hand platforms as an attractive consumer niche that retailers could target. Moreover, we advise policy makers to consider the impact of online second-hand transactions on economic, social, and environmental sustainability. Regarding the first, we find new evidence of digital divide: those on low incomes are less prone to transact in online second-hand markets, therefore limiting the prospects of the circular economy. Moreover, the participation of better-off consumers may suggest that second-hand transactions satisfy materialistic consumption which may not take place in the absence of such transactions. Thus, this would imply a smaller impact on the reduction of production and of brick and mortar stores, limiting the positive effects on the environment. From an academic perspective, whether second-hand consumption substitutes or complements first-hand consumption is an interesting avenue

of research. A first step to establish such conclusion would require learning about the types of goods transacted and the characteristics of the consumers involved in those transactions.

The remainder of this paper is organized as follows. In Section 2, the research hypotheses are laid out together with the literature review. Section 3 describes the database used and the econometric model. Section 4 contains the main results. In Section 5 these results are discussed. Finally, Section 6 concludes raising implications for retailers, policy-makers and researchers.

## 2. Literature Review and Hypothesis Development

This section draws our main hypotheses to explain consumer participation in online second-hand transactions, which are based on the previous literature findings. Following relevant preceding exercises [6], online second-hand shopping is likely to depend on multiple dimensions: economic reasons, situational factors, and individual characteristics (including socio-demographic variables and hedonistic attitudes such as the self-fulfillment of eco-friendly consumption).

### 2.1. Economic Variables

Experts consider that economic factors (financial benefits, cost awareness) are key drivers of CC [28]. Research on CC confirms the relevance of economic factors in explaining why individuals engage in this form of consumption [8], especially the aim of saving money [29] and their price-consciousness [30].

Specific research of second-hand transactions [10] finds that it is driven by practical reasons (chore shopping) rather than by recreational considerations (pleasure shopping). Consumers value the availability of branded goods at a low price [6,31], trying to get the best quality/price ratio [32] or other relative advantages [23]. Economic considerations seem especially important for buying second-hand durable goods, like furniture [33].

The perceived bargaining power [6] could be a differential factor vis-à-vis online first-hand shopping. While e-commerce is dominated by big retailers, second-hand is mostly P2P and buyers feel they have more room to get better prices.

In fact, the development of digital platforms to buy/sell goods may have attracted especially the most economically-driven, because of the swiftness to complete transactions, the wider range of opportunities to get a best deal and the easier comparability of prices [34].

As a result, we propose Hypothesis 1 (H1):

**Hypothesis 1 (H1).** *Consumer price-awareness fosters the use of second-hand online platforms.*

Income is another economic factor worth mentioning. Income could increase online expenditure in second-hand goods in some cases [34], since second-hand online platforms could attract wealthy consumers, in the form of materialistic and indulgent consumption [5]. Indeed, specific research on apparel goods [24] confirms that access to status goods (at an affordable cost) is very relevant to explain participation in CC schemes, e.g., through renting [22].

However, some surveys on the sharing economy in Spain have found it not significant to explain participation in this kind of platform [35]. Specific research on online fashion renting did not find income to be significant either [22,23].

Thus, we propose Hypothesis 2 (H2):

**Hypothesis 2 (H2).** *Consumers' income increases their use of second-hand online platforms.*

### 2.2. Situational Variables Affecting Consumers' Convenience

Situational factors include aspects like physical surroundings, temporal perspectives, or antecedent states [26,27]. They affect the consumer's decision without being related to personal attributes or to specific characteristics of the possible product choices. Situational

aspects are very relevant in the decision to buy goods using online technologies [36], because of factors like "time pressure" and "geographical distance".

These variables can be captured by the concept of "convenience," which managers/founder of P2P platforms consider key [37]. The consumers' willingness to save efforts/energy and time fosters the acquisition of second-hand goods online [6,30] and the use of renting platforms [17]. Convenience of second-hand marketplaces depends on the ability to access the goods [38] compared to the alternative: the availability and affordability of first-hand goods.

If the number of retail stores in the geographic area is low, e.g., owing to restrictive regulation entry (zoning) and exercise (opening times), stores would be more expensive and less accessible, leading consumers to turn to second-hand markets. In Spain, the 17 regional authorities establish their own regulation of the retail sector and local authorities have a role too through urban and zoning regulations, so there are important differences in store availability between administrative regions/provinces. This exerts an influence on the use of e-commerce [39] and, potentially, on the use of second-hand online platforms to satisfy consumer needs.

At the same time, one must bear in mind the particular functioning of second-hand platforms. Even if consumers' browsing and negotiation of prices takes place online, closing the transaction generally entails an actual face-to-face meeting where the potential buyer can actually check the state of the good and consumers actually tend to pay in cash when meeting [1]. This is why owning a car may increase the use of second-hand platforms.

Against this backdrop, we propose Hypotheses 3 (H3) and 4 (H4):

**Hypothesis 3 (H3).** *A low number of retail stores around the buyer raises the use of second-hand online platforms.*

**Hypothesis 4 (H4).** *Owning a car promotes the use of second-hand online platforms.*

Other related variables that could be considered are total population and density. Rural areas could use e-commerce more because the impact of transportation costs is lower than in the offline economy [40], thus facilitating the access to a wider supply of goods and services [39]. However, the role of these factors in second-hand online platforms may differ, because of two reasons. Firstly, the transaction typically entails an actual face-to-face meeting of the parties, in which case the incidence would decrease in sparsely populated areas. Secondly, the P2P nature of these markets involves a double matching of needs, which is more likely in highly populated areas. Therefore, higher population and density could imply "deeper" second-hand marketplaces (more potential and convenient transactions), reducing the "risk of scarcity" [18].

Hence, we propose Hypotheses 5 (H5) and 6 (H6):

**Hypothesis 5 (H5).** *Total population in the area increases the use of second-hand online platforms.*

**Hypothesis 6 (H6).** *Population density increases the use of second-hand online platforms.*

### 2.3. Individual Characteristics

Age and gender are socio-demographic factors worth mentioning. Although the effect of age is bound to depend on the goods purchased, we presume that generally the elderly will be less likely to engage in online second-hand markets. They can trust less (or stigmatize more) the idea of buying second-hand goods from unknowns through these webs/apps.

Regarding gender, some surveys on the use of sharing economy in Spain obtain that being a female is likely to increase participation [35], although these authors caution that this merits further research. Sectoral exercises in specific sub-segments of second-hand markets like clothing [9,23,34] obtain that women tend to be more present.

So, we propose Hypotheses 7 (H7) and 8 (H8):

**Hypothesis 7 (H7).** *Age has an effect on the use of second-hand online platforms.*

**Hypothesis 8 (H8).** *Gender may have an effect on the use of second-hand online platforms.*

Regarding the impact of the level of studies in online second-hand shopping, specific research on fashion renting did not find it significant [22,23]. According to experts on the sharing economy [28], education can be a driver of engagement in these platforms but especially when it is related to information technologies.

That is why we focus specifically on skills related to internet tools, using as a proxy the frequency of use. Spending time online can increase the likelihood of buying used products [34] through different channels.

Firstly, skilled and frequent internet users have more confidence in these platforms. Trust is a critical success factor for CC [19,34,37] and is pondered by platform managers and founders [37], especially when second-hand goods are involved [6,14]. Consumers who are not comfortable with online tools will tend to prefer personal sources to exchange second-hand goods [16], while frequent and experienced internet users are more likely to participate in online second-hand shopping [12,34]. The use of online tools, and specifically social networks, may capture also the socializing aspect, which (together with perceived convenience) is very relevant in some segments of the sharing economy like carpooling [29].

Secondly, internet-literate consumers are more likely to value the convenience and/or the quality of an app/website vis-à-vis face-to-face shopping [12,13,17,23,37,41].

The convenient use of these online platforms and their social dimension seem to have a differential impact on online second-hand trading (compared to offline). This is backed by an inquiry into resellers' motivations [12], but the same arguments could easily apply to buyers. The use of digital tools is crucial in online second-hand transactions [31].

Therefore, we propose Hypotheses 9 (H9) and 10 (H10):

**Hypothesis 9 (H9).** *The level of studies may increase the use of second-hand online platforms.*

**Hypothesis 10 (H10).** *More frequent use of internet increases the use of second-hand online platforms.*

Another factor to consider is the fact of having children because of the transitory need for special items [42]. Some children goods are relatively expensive and durable, and second-hand marketplaces offer an opportunity to buy these products (and resell them after some years too). This, coupled with parents' lack of time and the fact that product browsing can be done from home at any time of the day [36], can explain why having children may increase participation in online second-hand goods [9,43,44], even if hygiene and safety concerns could be a refraining factor for parents [44].

Thus, we propose Hypothesis 11 (H11):

**Hypothesis 11 (H11).** *Having children increases the use of second-hand online platforms.*

Finally, environmental concerns are a driver of participation in forms of the sharing economy and CC with a potential impact on sustainability, such as carpooling [29]. In the same vein, altruistic factors and the contribution to collective goods can be a motivation for engaging in second-hand transactions [31,45]. Some articles have found some significance of environmental factors in the specific case of renting fashion items online [23], circular economy of fashion products [46], or second-hand shopping of furniture [33]. But, as we said above, CC could also end up generating indulgent consumption and some general assessments of CC point to the fact that neither users [17,30] nor experts [28] deem environmental factors very relevant.

In any case, we propose Hypothesis 12 (H12):

**Hypothesis 12 (H12).** *Individuals' environmental awareness encourages the use of second-hand online platforms.*

Finally, Figure 1 provides the conceptual research model.

**Figure 1.** Conceptual research model.

## 3. Materials and Methods

### 3.1. Data

The source of the data used in this study is a survey ran by the Spanish National Markets and Competition Commission in the second quarter of 2017. The sample was designed to be representative of the population living in private households in Spain. The information was provided by 6705 individuals ≥18 years old, who used the internet at least once a week.

The interviewees were asked to provide information on whether they had bought or hired a second-hand good using online platforms such as eBay, Wallapop, Milanuncios or Segundamano/Vibbo in the previous 12 months. As shown by Table 1, a quarter of them had used these platforms in that period.

Table 1 provides the summary statistics for the variables used in the study. We use the labor market status as a proxy for income (with full-time employment pointing to more wealthy consumers compared to part-time, unemployed, etc.).

Some of the survey variables are interesting. The percentage of individuals with a solar panel in their home is 4.4%. We use this variable as a proxy for environmental consciousness. The percentage of individuals not knowing the type of electricity tariff in their household contracts is 37.2%. We consider not knowing the specific tariff type in one's home as a proxy of the individuals' lack of price-awareness, since electricity is a recurrent and important part of household expenditure, around 2–4% in Spain [47].

Some variables have been retrieved from other databases and incorporated to the main database. The number of retail stores in the individual's province, which excludes supermarkets and groceries, has been obtained from a survey on retail trade [48,49]. There are on average 10.6 retail shops per 1000 inhabitants. The population and the population density of the individuals' municipality have been extracted from the National Statistics Institute [50].

**Table 1.** Descriptive statistics.

| Variable | Obs | Mean | Std. Dev | Min | Max |
|---|---|---|---|---|---|
| Online second-hand purchase | 6705 | 0.248 | 0.432 | 0 | 1 |
| Age: 18–24 | 6705 | 0.085 | 0.279 | 0 | 1 |
| Age: 25–34 | 6705 | 0.090 | 0.287 | 0 | 1 |
| Age: 35–49 | 6705 | 0.319 | 0.466 | 0 | 1 |
| Age: 50–64 | 6705 | 0.360 | 0.480 | 0 | 1 |
| Age: 65 or over | 6705 | 0.145 | 0.352 | 0 | 1 |
| Gender: Female | 6705 | 0.586 | 0.493 | 0 | 1 |
| Employment: Full time | 6705 | 0.441 | 0.497 | 0 | 1 |
| Employment: Part time | 6705 | 0.078 | 0.268 | 0 | 1 |
| Employment: Retired | 6705 | 0.134 | 0.341 | 0 | 1 |
| Employment: Unemployed | 6705 | 0.162 | 0.368 | 0 | 1 |
| Employment: Student | 6705 | 0.101 | 0.302 | 0 | 1 |
| Employment: Non-employed | 6705 | 0.084 | 0.277 | 0 | 1 |
| Education: Primary | 6705 | 0.091 | 0.288 | 0 | 1 |
| Education: Secondary | 6705 | 0.594 | 0.491 | 0 | 1 |
| Education: University 3-years degree | 6705 | 0.152 | 0.359 | 0 | 1 |
| Education: University 5-years degree | 6705 | 0.163 | 0.369 | 0 | 1 |
| Children at home | 6705 | 0.241 | 0.428 | 0 | 1 |
| Population (thousands) | 6705 | 448.7 | 873.6 | 0.046 | 3142.0 |
| Density (thousands per km$^2$) | 6705 | 2.785 | 4.080 | 0.002 | 21.029 |
| Retail shops (per thousands of inhabitants) | 6705 | 10.666 | 1.197 | 7.380 | 13.997 |
| Solar panel | 6705 | 0.044 | 0.205 | 0 | 1 |
| Car | 6705 | 0.905 | 0.293 | 0 | 1 |
| Electricity tariff unknown | 6705 | 0.372 | 0.483 | 0 | 1 |
| Frequency of internet use: several times a day | 6705 | 0.677 | 0.468 | 0 | 1 |
| Frequency of internet use: almost every day | 6705 | 0.250 | 0.433 | 0 | 1 |
| Frequency of internet use: weekly | 6705 | 0.073 | 0.260 | 0 | 1 |

*3.2. Model Specification*

To identify the relationship between the economic, situational, and other individual characteristics and the likelihood of individuals buying or renting online second-hand goods in the previous year, a binary logit regression was fitted. Maximum likelihood estimation procedure was used to obtain the regression coefficients. In addition, the associated odds ratios were derived by exponentiating the previous coefficients.

**4. Results**

The results of the logit regression analysis are presented in Table 2, where the odds ratios, their standard errors and their 95% confidence intervals are reported for each variable.

First, we turn to the effects of personal characteristics on the participation in second-hand markets. Socio-demographics are important. Senior citizens are less likely to use online platforms to buy/rent second-hand goods, supporting (H7). For example, the odds of the older groups (from 50 to 64 and 65 or over years old) approximately halve those of the reference group (between 18 and 24 years old). Moreover, being a female has a

significant negative effect on participation in online second-hand markets: it reduces the odds of participation by approximately one fifth, supporting Hypothesis (H8).

**Table 2.** Logit model for online second-hand purchasing (N = 6705).

| Variable | Odds Ratio | Standard Error | 95% LCI | 95% UCI |
|---|---|---|---|---|
| Age: 18–24 (ref.) | 1 | | | |
| Age: 25–34 | 1.325 | 0.228 | 0.945 | 1.857 |
| Age: 35–49 | 0.829 | 0.156 | 0.573 | 1.199 |
| Age: 50–64 | 0.543 *** | 0.103 | 0.375 | 0.788 |
| Age: 65 or over | 0.409 *** | 0.099 | 0.254 | 0.659 |
| Gender: Female | 0.786 *** | 0.050 | 0.694 | 0.891 |
| Employment: Full time (ref.) | 1 | | | |
| Employment: Part time | 0.945 | 0.109 | 0.754 | 1.185 |
| Employment: Retired | 1.000 | 0.153 | 0.741 | 1.350 |
| Employment: Unemployed | 0.861 * | 0.076 | 0.725 | 1.024 |
| Employment: Student | 0.631 *** | 0.109 | 0.450 | 0.886 |
| Employment: Non-employed | 0.728 ** | 0.103 | 0.552 | 0.960 |
| Education: Primary (ref.) | 1 | | | |
| Education: Secondary | 1.090 | 0.125 | 0.870 | 1.364 |
| Education: University 3-years degree | 1.071 | 0.142 | 0.826 | 1.389 |
| Education: University 5-years degree | 1.226 | 0.162 | 0.946 | 1.589 |
| Children at home | 1.333 *** | 0.097 | 1.155 | 1.539 |
| Population (thousands) | 0.99997 | 0.00004 | 0.99989 | 1.00005 |
| Density (thousands per km$^2$) | 0.991 | 0.008 | 0.975 | 1.008 |
| Retail shops (per thousands of inhabitants) | 0.887 *** | 0.025 | 0.839 | 0.937 |
| Solar panel | 1.444 *** | 0.193 | 1.111 | 1.877 |
| Car | 1.423 *** | 0.168 | 1.129 | 1.795 |
| Electricity tariff unknown | 0.785 *** | 0.049 | 0.695 | 0.887 |
| Frequency of internet use: weekly (ref.) | 1 | | | |
| Frequency of internet use: almost every day | 1.805 *** | 0.323 | 1.271 | 2.564 |
| Frequency of internet use: several times a day | 3.283 *** | 0.560 | 2.349 | 4.587 |

* Significant at 10%. ** Significant at 5%. *** Significant at 1%. LCI: lower confidence interval. UCI: upper confidence interval.

Owning a car and having children are statistically significant and increase the odds of using second-hand platforms, (respectively) by 42.3% and 33.3%, supporting (H4) and (H11).

Coming to economic considerations, Table 2 shows that not knowing the type of electricity tariff has a significant effect and reduces the odds of second-hand participation by one fifth. Since this variable is a proxy for the price-awareness of individuals, we conclude that individuals who care less about prices and savings, participate less in online second-hand markets, as expressed in (H1).

Another economic variable worth mentioning is labor market status (which is used as a proxy for income). Being unemployed or a student (i.e., low earners) has a statistically significant negative effect on participation, diminishing its odds by 13.9% and 36.9% with respect to those in full-time employment, thus backing (H2).

More factors can be pointed out. The degree of familiarity with the internet increases the use of these platforms, confirming (H10). Compared to the odds of consumers who use

the internet weekly, consumers who use internet almost every day almost double the odds to use these second-hand apps, and consumers who use the internet several times a day more than triple the odds.

The odds ratio of individuals pertaining to households with solar panels is statistically significant and greater than one. This variable is a proxy for environmental awareness, so we conclude by supporting (H12): that the environmentally-concerned are more likely to engage in online second-hand markets.

Finally, increasing the availability of retail stores in the vicinity of the individual (at the level of the province) has a significant negative effect on their use of online platforms to buy/rent second-hand goods. This means that individuals living in provinces with a higher number of retail shops (per thousands of inhabitants), are less prone to engage in second-hand markets, confirming (H3). However, neither population (H5) nor density (H6) variables are statistically significant. Similarly, the odds ratios for the level of study variable are not statistically significant (H9).

## 5. Discussion

This study aims at fostering the understanding of why consumers choose to engage in online second-hand markets. Following the literature, we have concentrated on looking at several individual characteristics, economic variables and situational factors as potential explanations. As mentioned in the introduction, one of the merits of our study is that it is based on a representative sample of individuals, inclusive of those engaging and not engaging in online second-hand markets, which is critical to empirically assess the effects of the different variables on the participation decision.

Coming to the economic variables, our findings concerning the positive effect of price awareness on the use of online platforms to buy/rent second-hand goods are in line with previous literature. Consumers exhibit price-awareness, they are motivated by practical reasons, notably finding a better quality/price ratio [6,10,23,32–34].

Similarly, our results confirm that some low-earner groups are less likely to engage in online second-hand markets. This somewhat counter-intuitive result is consistent with other findings in the literature [34] and can be explained because of the reported characteristics of second-hand consumption: indulgent consumption [5], seeking for unique [51] or trendy goods [17], especially in apparel [22,24], children's equipment, toys, and video-games, etc.

Also confirmed by our results, having children can increase participation in these second-hand platforms, as has been suggested previously [9,43,44]. These platforms can be used to buy durable and relatively expensive second-hand goods which are needed temporarily (and that can be resold later on). The online dimension of these marketplaces (compared to the traditional offline channels for second-hand transactions) introduces rating, review and recommendation mechanisms which can give parents some assurances (e.g., regarding products hygiene, safety . . . ).

Situational factors, like the lack of availability of retail stores around and the ownership of a car (which facilitates the last stage of the transaction which usually takes place face-to-face), also increase the use of second-hand platforms. The previous literature had already emphasized the role of "convenience" [6,17,38].

Socio-demographic factors are relevant too. The participation in online second-hand markets decreases for senior citizens, despite the fact that previous exercises had not found age to be significant for the specific case of apparel and fashion products [22,23]. Regarding gender, we have obtained that being a male increases participation, opposite to previous results [35] and to the findings for a specific sub-segment like clothing [23,34].

Our paper also confirms that the frequency of internet use, which is related to internet literacy, also fosters participation in second-hand platforms, very much in line with previous results [34]. Frequent users of the internet have more trust in online tools, which is critical in second-hand goods [6,12,14,33,34,44,52,53]. Moreover, frequent internet users are more

capable of extracting the most of the apps/websites functionalities [12,13,17,23,37] and are more likely to enjoy the experience from an hedonic point of view [10,34,43,54].

Finally, we have found an explanatory role of an altruistic factor like environmental awareness, in line with some previous research [23,33,46,55], although there are also some exercises that had rejected this hypothesis for specific items like apparel [22].

## 6. Conclusions

The use of online platforms to buy or rent second-hand goods is a relevant phenomenon, both quantitatively and qualitatively. Quantitatively, users were almost 25% of internet users in Spain in our study and this figure is likely to have increased in the recent years. Qualitatively, some economic, situational, and socio-demographic variables stand out as explanatory factors, which has profound implications both for retailers and policy-makers.

As far as retailers are concerned, even if some (situational and socio-demographic) factors are beyond their control, they must be aware of this phenomenon and adapt to it [14,36,56]. According to our research, participants in online second-hand platforms are an attractive niche (in terms of willingness/ability to pay): full-time job, with children and car, eco-friendly, and spending time on the internet (although it is fair to say that these consumers are also cost-aware). Moreover, it seems that, differently from other areas of the online economy [36], consumer participation in online second-hand markets is driven by structural factors (e.g., price awareness, income, environmental awareness, internet skills) which do not change abruptly. Therefore, on the practical side, retailers can try to target this niche in order not to lose business opportunities [10,56].

It is important to bear in mind that, since agents seek specific products in second-hand platforms, brands may see their image affected. That is why some retailers (like Ikea, Fnac or Decathlon) have reacted to the phenomenon by rebuying/reselling used products of their own brand or by acting as a platform/matchmaker [57]. These can give potential buyers assurances regarding the quality of the product and of the vendor [14].

As far as policy-makers are concerned, they must consider the impact of this phenomenon on economic, environmental, and social sustainability.

On the positive side, this phenomenon may be attracting environmentally aware consumers. It allows the satisfaction of (temporary) needs in a circular economy, increasing economic and environmental efficiency. Besides, this phenomenon facilitates consumption smoothing for families with children, pointing to positive social derivatives too.

But we have seen that this phenomenon comprises especially well-off consumers (with full-time jobs) and who are relatively internet-literate. Therefore, if buying/renting goods through these online platforms allows costs savings, the fact that low-income consumers are not so active in these platforms undermines the contribution of this phenomenon to economic and social sustainability (pointing to more evidence of digital divide). In addition, the participation of relatively well-off consumers could indicate some traces of materialistic consumption (in apparel, children and sport equipment, video-games, collection items, etc.), pointing to a not-so-positive dimension in terms of economic, social, and environmental sustainability.

Furthermore, second-hand transactions revive the debate on whether the reduction in the production of first-hand goods (despite potential environmental benefits) is worse in economic and social terms because of its negative impact on GDP, employment creation, and tax collection.

Even from a strictly environmental perspective, the final impact is likely to depend on the type of goods exchanged. An increase in second-hand transactions, when involving durable energy-intensive goods (such as vehicles or domestic appliances), may imply a slower pace of adoption of more eco-friendly goods [56], although in general the impact of second-hand trading should be positive due to the avoided environmental costs caused by the production of new goods [30]. Nonetheless, it is important to recall that many transactions take place face-to-face [1], entailing an actual trip to exchange the good.

Therefore, this could actually increase car use, while a model of a more walkable city with brick-and-mortar stores could be friendlier from an environmental standpoint. This issue is relevant given that we have obtained that the lower availability of stores in an area increases the exchange of second-hand goods online. This debate informs the role to be played by horizontal (e.g., zoning) and sectoral regulation (e.g., retail trade entry and opening times).

Our findings are subject to some limitations, that point to potential directions of future research. To start with, our study focuses on second-hand purchases through apps or platforms. However, the phenomenon of second-hand shopping is broader as consumers can also buy second-hand products in online stores or in specialized brick and mortar shops. A recent study shows that in Spain among second-hand consumers apps are the most used option for transactions (72%), followed by shopping in online stores (52%) and in physical stores (42%) [58]. For future research it would be interesting to check whether the results found here for shopping via apps/platforms also apply to the other second-hand shopping modalities.

Moreover, a deeper understanding of some of the motives to participate in online second-hand markets is needed, to assess the role of economic, social, and environmental factors that affect the sustainability dimension. First, which types of goods are transacted and a sub-sectoral analysis of the drivers of consumer participation is relevant to understand whether first and second-hand consumption are substitutes or complements. Second, some factors can be cultural and country-specific, so replicating this exercise in other countries could help to check the validity of some conclusions. Some exercises that have compared the attitudes towards online second-hand shopping in different countries have obtained distinct factors depending on social customs and values [31].

Finally, the analysis of this question in the post-COVID scenario also merits thorough research. These types of transactions seem to have weathered relatively well throughout post-COVID economic uncertainty, at least in the case of clothing [3]. Although at the same time, hygiene and safety concerns could affect these dynamics.

**Author Contributions:** Conceptualization, P.H., B.G.-M. and D.S.; methodology, P.H., B.G.-M. and D.S.; software, P.H., B.G.-M. and D.S.; validation, P.H., B.G.-M. and D.S.; formal analysis, P.H., B.G.-M. and D.S.; investigation, P.H., B.G.-M. and D.S.; resources, P.H., B.G.-M. and D.S.; data curation, P.H., B.G.-M. and D.S.; writing—original draft preparation, P.H., B.G.-M. and D.S.; writing—review and editing, P.H., B.G.-M. and D.S.; visualization, P.H., B.G.-M. and D.S.; supervision, P.H., B.G.-M. and D.S.; Project administration, P.H., B.G.-M. and D.S. All authors have read and agreed to the published version of the manuscript.

**Funding:** This research received no external funding.

**Institutional Review Board Statement:** Not applicable.

**Informed Consent Statement:** Not applicable.

**Data Availability Statement:** Not applicable.

**Conflicts of Interest:** The authors declare no conflict of interest.

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
