# Peer review of "Drivers of Consumer Participation in Online Second-Hand Transactions†"

_sustainability, doi:10.3390/su14074318_

Round 1

Reviewer 1 Report

The article is actual and it really advances knowledge in the field of second-hand consumption. The authors clearly state the points through which their research contributes to the study of second-hand consumption. The literature is actual and relevant, the hypotheses are correct, the analysis is also correct and the same is the case with the conclusions. I highly appreciate the voluminous dataset.

I have two critical remarks:

  1. I think that there is a need to discuss the case of online second-hand shopping also in the direction of formal second-hand shops. Platforms are one thing, they are between formal and informal, but there are formal online second hand shops as well. Do you have previous data in this direction? How the results obtained relate to them? I think there is a need for such a discussion.
  2. I do not insist, but a short review on second-hand consumption from Spain in general would be welcome. It could help to put the results in a comprehensive context.

Author Response

Dear Reviewer 

Thanks very much for the comments that have helped to improve the manuscript. Apart from including the reponses to your comments we also include in the attached file the responses to the other reviewers so that you can understand the explanation for the other changes in the manuscript.

In order to identify your comments you are labelled as Reviewer 1

Reviewer 2 Report

Overall a very interesting topic; while the topic is indeed of great value to both academia and the practitioner communities, there remain a few issues that ought to be addressed before this paper can be accepted.
I have only a few concerns about the paper and some suggestions that maybe the authors could consider:
1.    In the 'Introduction' section, the proposed research gap and the stated objectives do not meet the criteria of proper synergy. Please make the research gap and the research objectives consistent with each other.
2.    This raises some concerns regarding the potential overlap with the authors' previous works. The authors should explicitly state the novel contribution of this work and its similarities and differences with their previous publications.
3.    The authors need to clearly articulate the academic and practical implications of this study. I would suggest writing a paragraph in the conclusion section for the implications. Also, state a few of the key implications at the end of the 'Introduction' section.
4.    I suggest some studies can enrich the literature of the study (DOI: https://doi.org/10.3390/su13116488; DOI: https://doi.org/10.1080/23311975.2021.2016556; DOI: https://doi.org/10.1080/23311975.2021.1978620). I think this aspect could reinforce the authors' literature and discussion.
5.    For readers to quickly catch your contribution, it would be better to highlight major difficulties and challenges and your original achievements to overcome them in a clearer way in the abstract and introduction.
6.    How could/should futures studies improve the model?
If these revisions can be made in the manuscript, I believe that this study can be accepted for publication.
I wish the authors all the very best with this study. 

Author Response

Dear Reviewer 

Thanks very much for the comments that have helped to improve the manuscript. Apart from including the reponses to your comments we also include in the attached file the responses to the other reviewers so that you can understand the explanation for the other changes in the manuscript.

In order to identify your comments you are labelled as Reviewer 2

Reviewer 3 Report

This manuscript investigated the role of economic variables, situational factors and individual characteristics based on a logit model to a sample of 6,705 internet users in Spain. Manuscript entitled "Drivers of consumer participation in online second-hand transactions" of interest to "Sustainability" journal. Presented study opens up prospects in this field of science. At the same time the separate suggestions are given as follows.

  1. In the second section, the research hypotheses are laid out together with the literature review. Usually, referring to the results of previous researches and the hypotheses that have been developed, then research model is created. However, the conceptual research model is not presented.
  2. The third section describes the database used and the econometric model. If possible, it is necessary to provide screenshots of the databases used.
  3. Finally, the sixth section concludes raising implications for retailers, policy-makers and researchers. This section is not very concise.

Author Response

Dear Reviewer 

Thanks very much for the comments that have helped to improve the manuscript. Apart from including the reponses to your comments we also include in the attached file the responses to the other reviewers so that you can understand the explanation for the other changes in the manuscript.

In order to identify your comments you are labelled as Reviewer 3
